# Nitrogen Metabolism during Anaerobic Fermentation of Actual Food Waste under Different pH Conditions

Chuyun Zhao [1,†], Luxin Yang [1,†], Huan Li [1,*] and Zhou Deng [2]

1 Shenzhen International Graduate School, Tsinghua University, Shenzhen 518055, China; 18817295185@163.com (C.Z.); ylx_hz@163.com (L.Y.)
2 Shenzhen Lisai Environmental Technology Co., Ltd., Shenzhen 518055, China; dengzhou2005@126.com
* Correspondence: li.huan@sz.tsinghua.edu.cn
† These authors contributed equally to this work.

**Abstract:** Acidogenic fermentation can convert food waste (FW) into small molecules of acids and alcohols, and the broth can be used as a carbon source of denitrification in wastewater treatment plants. However, the soluble nitrogen-containing substances generated in fermentation influence the quality of the carbon source, and microbial nitrogen transformation under different pH conditions has rarely been reported. In this study, four FW fermentation systems were operated continuously with a focus on nitrogen transformation, and metagenomic and metatranscriptomic analyses were used to reveal the metabolic pathways. The results showed that approximately 70% of nitrogen existed in solid organic matter, and the dissolution of solid proteins was limited at pH 4.0–5.0. The concentration of soluble nitrogen, encompassing both soluble organic nitrogen and ammonium, remained relatively stable across various pH conditions. However, high pH values promoted the conversion of soluble nitrogen-containing substances to ammonium, and its concentration increased by 122%, 180%, 202%, and 267% at pH 4.00, pH 4.27, pH 4.50, and pH 5.00. *Lactobacillus* played a crucial role in ammonium production via the arginine deiminase pathway at pH 4.0–4.5, and *Prevotella* was the key contributor with the assistance of glutamate dehydrogenase at pH 5.0. The findings provide insights into organic nitrogen transformation in acidogenic fermentation for optimizing FW treatment processes.

**Keywords:** food waste; fermentation; protein; amino acid; metabolism

## 1. Introduction

Food waste (FW) is widely generated during food processing, transportation, storage, and consumption. FW is also a significant component of municipal solid waste (MSW). In China, FW accounts for more than 50% of MSW, i.e., 130 million tons of FW is produced annually [1]. FW is rich in organic matter and nutrients, with carbohydrates and proteins as the primary constituents, making up over 70% of the total solid (TS) content [2,3]. Consequently, improper treatment could result in secondary pollution and significant greenhouse gas emissions. In contrast, FW can be utilized through different pathways to produce biogas for energy recovery, fertilizer, animal feed, and chemicals, such as ethanol, acetic acid, propionic acid, etc.

Anaerobic digestion is a commonly used treatment technology [1]. It can convert the organic matter in FW into biogas, usually used to generate heat or electricity [4]. A two-phase anaerobic digestion process is usually adopted to avoid excessive acidification and inhibit methanogenesis. FW is hydrolyzed and decomposed into micromolecular organic acids and alcohols in the first acid-producing stage. In the second methanogenic phase, the acids and alcohols are further degraded into acetic acid and hydrogen, which methanogens use to generate methane. In some cases, the first phase can be used alone, with small molecules of organic acids or alcohols as the target products. This process is called fermentation [5,6]. Such a process allows for the maximization of organic matter

exploitation and offers other advantages, such as a short treatment cycle, high tolerance to shock loads, and a small footprint [7]. Carbohydrates are the primary substrate in producing acids or alcohols. Proteins can be degraded into amino acids, and only a tiny part is used for cell building. The carboxyl groups are converted into organic acids for most amino acids, while the amino group finally becomes ammonia or ammonium. Therefore, some nitrogen-containing substances are present in addition to organic acids and alcohols in the fermentation liquid.

Fermentation liquid containing acetic acid, propionic acid, ethanol, etc. can be used as the carbon source for biological denitrification in wastewater treatment plants (WWTPs) [8,9]. Methanol, ethanol, and glucose are the typical commercial carbon sources [8]. The combination of FW fermentation and wastewater treatment provides a new pathway for FW utilization with high efficiency and low cost. Also, it reduces the economic and environmental burden of WWTPs by avoiding the usage of fossil carbon sources [1]. However, the presence of soluble nitrogen-containing compounds influences the quality of fermentation liquid as a carbon source [10–12] and consequently challenges the standard discharge of WWTPs. Therefore, this new pathway of FW utilization must disclose the nitrogen transformation first and finally reduce the dissolution of solid nitrogen by controlling FW fermentation conditions.

Nitrogen transformation is crucial in determining the nitrogen distribution in fermentation liquid [13,14]. Most previous studies focused on converting organic matter fully in fermentation and separating fermentation products. Meanwhile, the conversion of nitrogen and its mechanism have yet to be studied. Some studies have measured the concentration of nitrogen-containing compounds in the system, such as TN, soluble protein, ammonium, etc. However, they have not analyzed the influence of fermentation conditions on nitrogen metabolism, which is essential for using fermentation liquid as a carbon source. Yin et al. mapped out protein conversion pathways during fermentation [15], and Shen et al. examined nitrogen distribution and metabolism in the fermentation of two protein-rich substrates [16]. However, the nitrogen metabolism under different conditions in FW fermentation has yet to be analyzed systematically, especially using actual FW as the substrate.

To fill the knowledge gap, we investigated nitrogen metabolism at different pH values during FW fermentation, corresponding to several common fermentation types in practice. Four mesophilic fermentation reactors were operated semi-continuously using actual FW slurry collected from a treatment plant as the substrate. Various nitrogen-containing substances were measured to establish nitrogen distribution in the systems. In this paper, we use metagenomic and metatranscriptomic analyses to characterize the microbial community and show the expression of different genes involved in the various steps of nitrogen metabolism. This study represents the first comprehensive investigation of nitrogen conversion and distribution during food waste fermentation under varying pH conditions, particularly emphasizing nitrogen distribution between the solid and liquid phases. The results of this study will enhance the understanding of nitrogen metabolism in FW fermentation and provide insights for controlling protein solubilization and ammonium release.

## 2. Materials and Methods

### 2.1. Substrate and Inoculum

The substrate and the inoculum used in the fermentation experiments were collected from a treatment plant in Shenzhen, China. The food waste treated by the plant is sourced from restaurants and households. It comprises waste staples, vegetables, meat, and other constituents. This plant adopted a conventional two-phase anaerobic digestion process to treat 500 tons of FW daily. During this process, the organic matter in FW was converted into biogas, which was used for electricity generation and heat recovery. Upon arrival at the plant, the raw FW underwent several treatment steps, including crushing, homogenization, acidogenic fermentation, three-phase separation, and anaerobic digestion. The homogenized FW slurry, with a particle size of approximately 5 mm, was collected every two weeks to serve as the substrate in the fermentation experiments. After collection, the substrate

was immediately transferred to the laboratory, where it was crushed to a size of less than 2 mm and stored at −20 °C until use. Due to variations in the raw FW composition, the characteristics of the substrate fluctuated. The biogas slurry discharged from the plant was taken as the inoculum in the fermentation experiments. The characteristics of the substrate and inoculums are shown in Table 1.

**Table 1.** Characteristics of the substrate and the inoculum.

| Item | Substrate | Inoculum |
|---|---|---|
| Total solid (TS, %) | 12.2 ± 1.2 | 1.9 ± 0.2 |
| Volatile solid (VS, %) | 11.6 ± 1.2 | 0.9 ± 0.3 |
| Soluble chemical oxygen demand (SCOD, mg/L) | 34,011 ± 5447 | 2242 ± 114 |
| Total chemical oxygen demand (TCOD, mg/kg) | 129,109 ± 12,791 | 7824 ± 857 |
| C/N | 15.3 ± 0.3 | 7.3 ± 0.3 |
| pH | 4.0~4.5 | 7.9 ± 0.2 |
| Oxidation reduction potential (ORP, mV) | / | −429 ± 15 |
| $NH_4^+$-N (mg-N/L) | 104.3 ± 36.9 | 3257 ± 473 |
| Total Kjeldahl nitrogen (TKN, mg-N/L) | 2240.8 ± 345.9 | / |
| Soluble Kjeldahl nitrogen (SKN, mg-N/L) | 662.4 ± 85.5 | / |
| Alkalinity (mg $CaCO_3$/L) | / | 6657 ± 215 |

## 2.2. Semi-Continuous Fermentation Experiments

Four continuously stirred tank reactors, each having a working volume of 1.2 L, were employed in conducting the fermentation experiments. Every reactor was furnished with an automated stirrer, a temperature regulator, and a pH controller. The stirring speed was set at 200 rpm, and an external circulating water bath maintained the temperature at 36 ± 1 °C. The pH levels in the reactors were regulated utilizing a pH meter alongside peristaltic pumps capable of introducing NaOH solution (5.0 mol/L) or HCl solution (2.0 mol/L) into the reactors. One reactor was designated as the control without pH control (R0), while the pH values of the other three reactors were automatically maintained at 4.00 ± 0.04 (R1), 4.50 ± 0.04 (R2), and 5.00 ± 0.04 (R3), respectively. The pH in R0 was finally stable at approximately 4.3. The pH conditions can determine the fermentation performance, and the pH range of 4.0–5.0 was suitable for harvesting small molecules of organic acids and alcohols [17].

At the start of the experiments, each reactor was loaded with 1080 mL of prepared FW as the substrate and 120 mL of biogas slurry as the inoculum. Subsequently, the reactors were run in a semi-continuous mode with a hydraulic retention time (HRT) of 4 days. On a daily basis, 300 mL of fermentation broth was substituted with fresh FW slurry. The discharged effluent was collected every two days for further analysis. Based on the substrate properties, the organic loading rate (OLR) ranged from 23 to 36 g VS/(L·d), which simulated the actual operating situation in a real-world treatment plant. The system achieved stability when the concentrations of the primary products deviated by less than 20% for a minimum of two hydraulic retention times (HRTs), as previously described [17]. Data collected during the stable period were used for systematic analysis.

## 2.3. Microbial Analysis

On the final day of the experiment, the fermentation broth was collected in duplicate from the reactors. The samples were swiftly frozen using liquid nitrogen and subsequently stored in a liquid nitrogen tank to maintain their integrity. Subsequently, the frozen samples were sent to Majorbio Inc. in Shanghai, China, for further extraction and sequencing processes.

The complete DNA and RNA were isolated using the Soil RNA Extraction Kit (Majorbio, Shanghai, China) and Soil RNA Extraction Kit (Majorbio, Shanghai, China), following

the respective manufacturer's instructions. The concentration and purity of the extracted DNA were assessed using TBS-380 (YPH-Bio, Beijing, China) and NanoDrop2000 spectrophotometers (Thermo Scientific, Waltham, MA, USA), respectively. Additionally, the quality of the DNA extract was verified using a 1% agarose gel. The extracted RNA's integrity and quantity were assessed using a NanoDrop 2000 spectrophotometer and an Agilent 5300 Bioanalyzer (Agilent Technologies, Palo Alto, CA, USA). The DNA paired-end library was prepared utilizing the NEXTFLEX® Rapid DNA-Seq (Bioo Scientific, Austin, TX, USA). The RNA was subjected to standard Illumina library preparation with an Illumina® Stranded mRNA Prep, Ligation (Illumina, San Diego, CA, USA), and rRNA was depleted using a RiboCop rRNA Depletion Kit for Mixed Bacterial Samples (Lexogen, Greenland, NH, USA). The DNA and RNA libraries underwent sequencing on an Illumina NovaSeq 6000 platform, resulting in the production of 150 bp paired-end reads. DNA and RNA sequencing data were more than 25 Gbp and 15 Gbp, respectively.

Quantitative PCR (qPCR) analysis was performed using an ABI7300 instrument (Applied Biosystems, San Francisco, CA, USA). ChamQ SYBR Color qPCR Master Mix ($2\times$) (Vazyme, Nanjing, China) was used with the 341F and 806R mix primer sets. Given the overwhelming dominance of bacteria (>99.9%) in our samples, the use of the primer pair 341F and 806R had negligible impact on the quantification of microorganisms. The PCR reactions consisted of a pre-heat start (95 °C for 3 min) and subsequent 40 amplification cycles (95 °C for 5 s, 58 °C for 30 s, and 72 °C for 60 s). After the reaction, a melting curve analysis was conducted to check the specificity.

### 2.4. Other Analytical Procedures

TS and VS were assessed through weight measurements subsequent to drying at 105 °C and incineration at 600 °C, correspondingly. The determination of TCOD and SCOD was carried out using COD digestion vials (Hach, Loveland, CO, USA) and a DR3900 spectrophotometer (Hach, USA). pH values and ORP values were obtained utilizing a digital meter (PHS–3C, INESA, Shanghai, China) paired with a pH probe (501, INESA, China) and an ORP probe (501, INESA, China). In order to analyze the soluble parameters, the samples were first centrifuged at $5800\times g$ for 5 min, followed by filtration of the resulting supernatant through a 0.45 μm filtration membrane. Subsequently, the obtained filtrate was utilized for the assessment of soluble parameters, encompassing SCOD, SKN, ammonium, and the fermentation products. TKN was determined utilizing the Kjeldahl method employing an automated Kjeldahl apparatus (Kjeltec 8200, FOSS, Hilleroed, Denmark), with column, injector, and detector temperatures maintained at 180 °C, 230 °C, and 250 °C, respectively. For SKN analysis, a Simplified TKN TNTplus Vial (TNT880, Hach, USA) was employed. A gas chromatograph (GC-2014, Shimadzu, Kyoto, Japan) outfitted with a capillary column (DB-FFAP 30 m $\times$ 0.320 mm $\times$ 1.00 μm) and a flame ionization detector was utilized to quantify alcohols such as ethanol and n-propanol, along with volatile fatty acids (VFAs) comprising acetic acid, propionic acid, n-butyric acid, n-valeric acid, n-caproic acid, and n-heptanoic acid. Lactic acid quantification was performed using a high-performance liquid chromatograph (LC-2030C 3D Plus, Shimadzu, Japan) featuring an InertSustain® C18 column (5 μm, 25 cm $\times$ 4.6 mm) and detection through an ultraviolet (206 nm) detector. The eluent used was 85% 20 mmol/L $KH_2PO_4$ and 15% methanol, with a 0.7 mL/min flow rate.

The statistical analysis was carried out employing IBM SPSS software (v29.0.1.0). A significance level of 5% ($p < 0.05$) was considered as statistically significant.

## 3. Results

### 3.1. Fermentation Performance

The substrate sourced from the FW treatment plant was used to simulate industrial operation conditions. The reactors (R0, R1, R2, and R3) were operated for 78 days, 64 days, 84 days, and 64 days, respectively. Daily data in the experiment are recorded in Figure 1, and the average values with their standard deviations during the stable stages

are summarized in Table 2. The initial concentration of lactic acid in the substrate was 9002 ± 4280 mg/L, constituting approximately 28% and 7% of the influent's SCOD and TCOD, respectively, suggesting that the FW slurry from the treatment plant had already undergone lactic acid fermentation to some degree at its collection, transportation, storage, and pretreatment stages. Initially, all the reactors continued lactic acid fermentation, with lactic acid concentrations peaking at around 20,000 mg/L on the second day. However, the lactic acid was rapidly consumed, and notable differences in fermentation types and product profiles emerged under varying pH conditions.

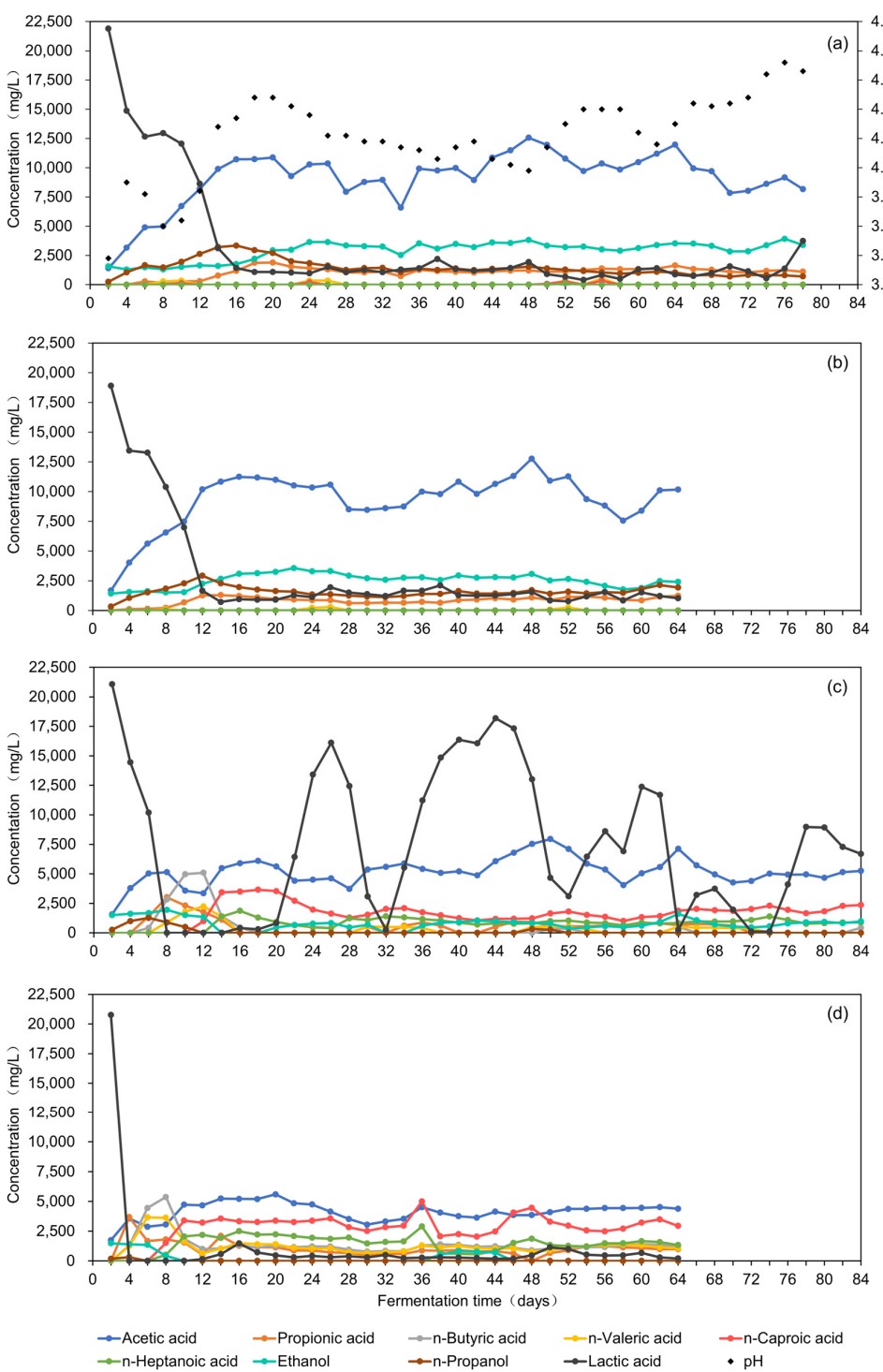

**Figure 1.** Variation of fermentation products during the experiment; (**a**) R0; (**b**) R1; (**c**) R2; and (**d**) R3.

**Table 2.** Fermentation products at the stable stage under different pH conditions.

|  | R0 | R1 | R2 | R3 |
|---|---|---|---|---|
| pH | $4.27 \pm 0.06$ | $4.00 \pm 0.04$ | $4.50 \pm 0.04$ | $5.00 \pm 0.04$ |
| Lactic acid (mg/L) | $1444 \pm 990$ | $1225 \pm 254$ | $7969 \pm 1005$ * | $522 \pm 243$ |
| Acetic acid (mg/L) | $8780 \pm 773$ | $9066 \pm 926$ | $5002 \pm 224$ | $4425 \pm 51$ |
| Propionic acid (mg/L) | $1185 \pm 99$ | $1061 \pm 146$ | 0 | $1080 \pm 120$ |
| n-Butyric acid (mg/L) | 0 | 0 | $108 \pm 188$ | $1240 \pm 114$ |
| n-Valeric acid (mg/L) | 0 | 0 | 0 | $1235 \pm 105$ |
| n-Caproic acid (mg/L) | 0 | 0 | $2029 \pm 299$ | $2912 \pm 332$ |
| n-Heptanoic acid (mg/L) | 0 | 0 | $840 \pm 46$ | $1430 \pm 154$ |
| Ethanol (mg/L) | $3312 \pm 350$ | $2167 \pm 274$ | $905 \pm 54$ | 0 |
| n-Propanol (mg/L) | $778 \pm 53$ | $1725 \pm 254$ | 0 | 0 |

* The average result at the end of the experiment.

The reactor without pH control (R0) and the reactor maintained at pH 4.0 (R1) exhibited similar performance. In both cases, lactic acid was replaced by acetic acid, n-propanol, ethanol, and propionic acid. After this transition, the reactors ran relatively steadily. In the case of R0, the pH value increased from 3.69 on the second day to more than 4.00 after the fourteenth day, finally stabilizing at 4.20–4.40. At the stable stage, acetic acid was the predominant product, with a concentration of $8780 \pm 773$ mg/L in R0 and $9066 \pm 926$ mg/L in R1. Additionally, ethanol, lactic acid, and propanol were also produced. However, carboxylates with more than three carbon atoms were barely detectable in both systems. In contrast, prior research suggested that the FW fermentation systems without pH control naturally evolved into lactic acid fermentation [17–19]. The unique result in this study may stem from the differences in substrates. Previous studies usually employed fresh FW as the substrate, implying the FW did not experience any biochemical reactions. In this study, actual FW slurry was collected from an FW treatment plant, and the substrate had undergone lactic acid fermentation to some extent before entering the fermentation reactors.

R2 (pH 4.5) exhibited a dramatic fluctuation in its product spectrum throughout the experiment, with lactic acid concentration swinging between a low level (100 mg/L) and a high level (20,000 mg/L). On average, lactic acid was the dominant product, followed by acetic acid ($5002 \pm 224$ mg/L). Notably, medium-chain fatty acids (MCFAs), particularly C6 and C7 carboxylates, were detected in the effluent, with concentrations reaching $2029 \pm 299$ mg/L and $840 \pm 46$ mg/L, respectively, during the stable stage. These findings align with the previous report [20], who also observed the fermentation system fluctuating at pH 4.5–4.7. Consequently, we hypothesize that this pH range may serve as a transitional boundary between different fermentation pathways. Hence, the pH change resulted in the shift of microbial metabolism, and the corresponding change of products (like lactic acid) caused a pH adjustment in reverse. This periodic variation was further analyzed in the following section.

The system displayed more stability when the pH was set at 5.0 (R3). The system tended to favor chain elongation reactions, leading to the production of more MCFAs. During the stable phase, the concentrations of C5, C6, and C7 carboxylates reached $1235 \pm 105$ mg/L, $2912 \pm 332$ mg/L, and $1430 \pm 154$ mg/L, respectively. Meanwhile, lactic acid, ethanol, and n-propanol concentrations were significantly reduced, suggesting they were likely consumed as electron donors for MCFA production. Similarly, lactic acid can be used as a platform chemical compound to biosynthesize other chemicals [21].

*3.2. Nitrogen Transformation*

Apart from water, FW primarily consists of carbohydrates, proteins, and lipids in the solid phase. Nitrogen elements in FW are primarily derived from proteins. The nitrogen

in FW can be categorized into insoluble nitrogen and soluble nitrogen. Insoluble nitrogen is the nitrogen within particles, comprising insoluble protein nitrogen and microbial cell nitrogen. Meanwhile, soluble nitrogen is the nitrogen dissolved in the fermentation liquid, which can be further divided into soluble organic nitrogen and ammonium. Nitrate and nitrite were not detected in the analysis, and thus, they will not be discussed further. Insoluble protein nitrogen, soluble organic nitrogen, and ammonium were determined through direct measurements, while microbial cell nitrogen was estimated based on the results from qPCR analysis.

Figure 2 presents a summary of the nitrogen distribution in the influent and the effluent of the fermenters. In the influent, approximately 70% of nitrogen existed in solid organic matter, and the majority (>99% by mass) of insoluble organic nitrogen was undissolved protein nitrogen (1570–1700 mg/L). The soluble organic nitrogen was 622.4 ± 85.5 mg/L, and the ammonium concentration was only 104.3 ± 36.9 mg/L. After fermentation, the average concentration of insoluble organic nitrogen showed insignificant differences ($p > 0.1$) in these reactors compared to the influent. The quantities of bacteria in R0, R1, R2, and R3 were approximately $4.44 \times 10^8$ copies/mL, $1.55 \times 10^8$ copies/mL, $6.28 \times 10^8$ copies/mL, and $9.68 \times 10^8$ copies/mL, respectively (Figure S1). It was assumed that the bacteria in the fermentation broth had the same weight as *Escherichia coli* at $2.8 \times 10^{-13}$ g per cell, and the water content of the cells was 75% [22]. The nitrogen content of the dry cells was 13%, according to the composition measurement of mixed anaerobic culture [23]. The cell nitrogen was finally calculated as 4.04 mg/L, 1.42 mg/L, 5.71 mg/L, and 8.81 mg/L, respectively. Although the higher pH relieved the inhibition and promoted the proliferation of microorganisms [24], the assimilation of nitrogen was minimal, with less than 10 mg/L of cell nitrogen, due to the limited energy obtained from anaerobic reactions. The solubilization of solid proteins was also limited in the fermentation process at pH 4.0–5.0. However, a portion of soluble organic nitrogen was utilized by microorganisms and converted into ammonium. The concentration of ammonium showed a positive correlation with pH conditions. At pH 4.0, uncontrolled pH (pH 4.2–4.4), pH 4.5, and pH 5.0, the ammonium concentration increased by 122%, 180%, 202%, and 267%, respectively, compared to the initial concentration in the influent. This result suggests that more active microorganisms utilized soluble organic nitrogen and released ammonium at higher pH levels.

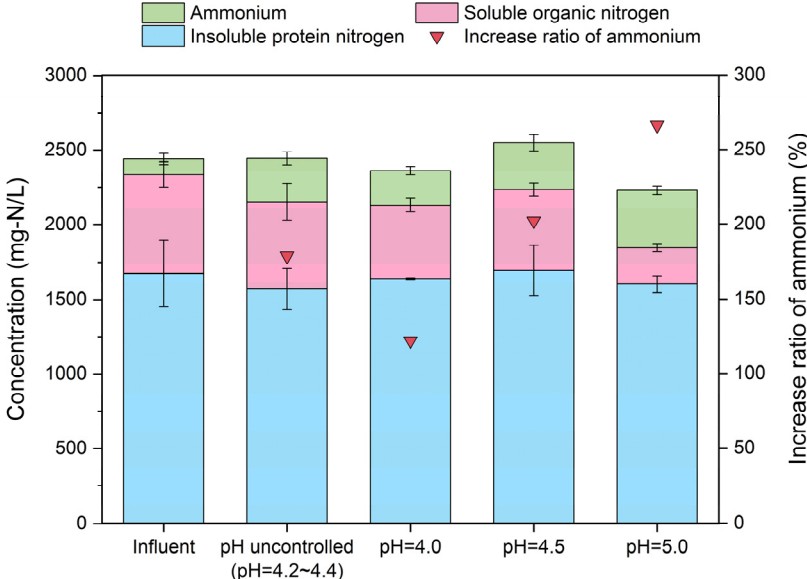

**Figure 2.** The nitrogen distribution in the different fermentation systems.

### 3.3. Microbial Community

Taxonomic annotation was performed based on metagenomic data. A total of 153 phyla, 280 classes, 606 orders, and 1311 families were identified in the four reactors. Firmicutes

emerged as the dominant phylum in R0, R1, and R2, accounting for 57%, 72%, and 63% of the total abundance, respectively, followed by Actinobacteria (40% in R0, 23% in R1, and 34% in R2). In contrast, Actinobacteria (37%) was the predominant phylum in R3, followed by Bacteroidota (33%) and Firmicutes (27%). The composition of the microbial communities at the family level is illustrated in Figure 3a.

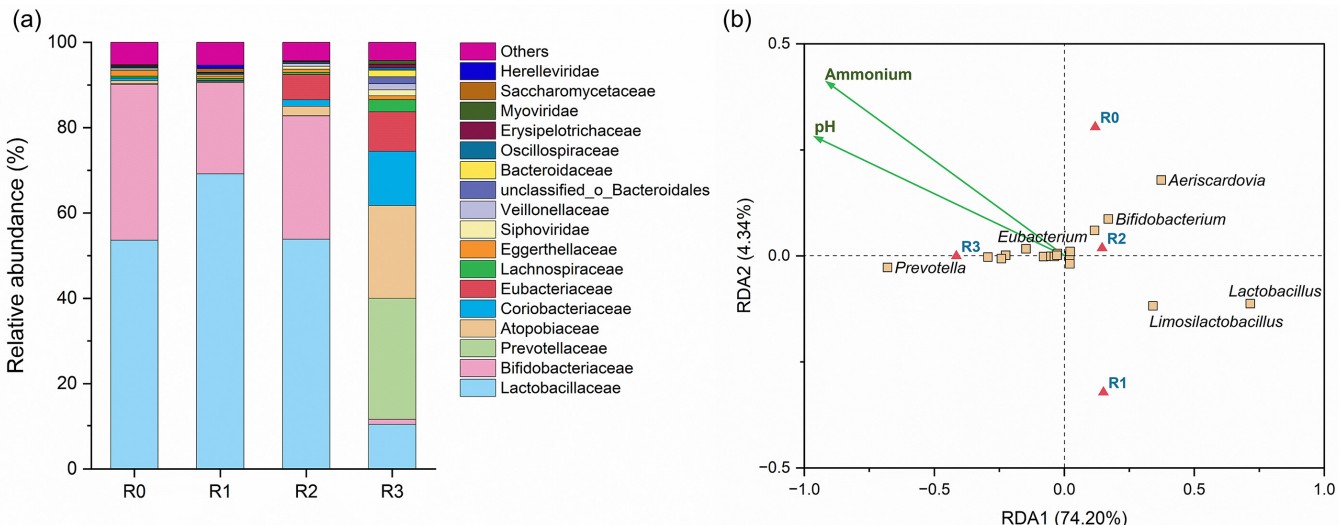

**Figure 3.** Microbial communities in fermentation reactors at different pH values; (**a**) relative abundances of the dominant taxa in the metagenome at the family level, and only the families with a relative abundance higher than 0.5% in DNA library are listed; (**b**) redundancy analysis to investigate the relationship between pH, ammonium concentration, and bacterial community.

Lactobacillaceae emerged as the dominant family in R0, R1, and R2, with its relative abundance ranging from 50% to 70%. Following it, the family Bifidobacteriaceae also displayed substantial relative abundance. A highly acidic environment can inhibit Bifidobacteriaceae [17], leading to its lower relative abundance in R1 (21.3%) compared to those in R0 (36.6%) and R2 (28.9%). The microorganisms belonging to Lactobacillaceae and Bifidobacteriaceae are typified as lactic acid bacteria (LAB) [20], capable of synthesizing lactic acid as a product. This result verified that the pH lower than 4.5 was highly conducive to lactic acid production. However, in the reactors R0 and R1, the lactic acid produced was further converted to ethanol, n-propanol, acetic acid, and propionic acid by specific microbial activities. Notably, the family Eubacteriaceae (5.9%), emerged in R2, are known as MCFAs producers that metabolize lactic acid into C6 and C8 carboxylates [25]. The proliferation of Eubacteriaceae promoted the consumption of lactic acid and the accumulation of n-caproic acid (Table 2), and it also explained the fluctuation of lactic acid concentration in R2. In R3, the microorganism displayed higher diversity. The relative abundance of LAB was only 11.4%, and the family Prevotellaceae (28.5%) became predominant, followed by Atopobiaceae (21.7%) and Coriobacteriaceae (12.7%). The Prevotellaceae and Atopobiaceae are known as VFA producers in mildly acidic environments [6], while the bacteria of the family Coriobacteriaceae can metabolite carbohydrates to produce lactate and acetate [25]. The proliferation of these bacteria promoted the conversion of carbohydrates to VFAs. In addition, Eubacteriaceae also proliferated in R3 with a relative abundance of 9.3%.

A minor fraction of LAB has proteolytic activity mediated by cell-envelope proteinases (CEPs) [26]. However, the prevalence of such LAB strains in the system was minimal. For instance, *Lactobacillus helveticus* comprised only approximately 2% of the microbial community in R0, R1, and R2. Other notable LAB species like *L. paracasei*, *L. rhamnosus*, and *Streptococcus thermophilus* were scarcely detected. Therefore, solid protein was rarely dissolved. LAB possess amino acid metabolic capabilities [27], and accordingly some ammonium was released from ammo acids at low pH levels. Some microbes from the families Prevotellaceae, Atopobiaceae, Coriobacteriaceae, and Eubacteriaceae are known for

their protease secretion and amino acid metabolism [28], but their capabilities are distinct significantly at the genus or species levels. We employed statistical methods to identify the essential microorganisms contributing to protein degradation, especially ammonium release. Redundancy Analysis (RDA) was used to visualize microbial structures at the genus level with varying environmental factors, as shown in Figure 3b. The analysis indicated that in this experiment, genera like *Prevotella* and *Eubacterium* were positively correlated with elevated ammonium concentration, with *Prevotella* showing a solid association.

### 3.4. Key Enzymes and Transporters

Microorganisms can secrete enzymes to break proteins into peptides and amino acids, which are then taken up by the membrane transport system and ultimately participate in metabolism [26]. Enzymes with proteolytic abilities are peptidases [29]. In the KEGG database, peptidases are further categorized into endopeptidases and exopeptidases. Endopeptidases (EC 3.4.21–3.4.99) cleave peptide bonds within the polypeptide chain at specific sites, while exopeptidases (EC 3.4.11–3.4.19) catalyze the cleavage of amino acids from either the N-terminus or C-terminus of the polypeptide substrate, resulting in the production of free amino acids and oligopeptides [30]. The expression of the two peptidases in different reactors is depicted in Figure 4a. The four reactors showed similar transcriptomic activities regarding endopeptidases, but the expression of exopeptidases exhibited a positive correlation with pH values. At uncontrolled pH, pH 4.5, and pH 5.0, the expression levels were approximately 3.6, 7.3, and 8.2 times higher than at pH 4.0. The results indicated that the reactors had similar abilities to break down protein chains at internal sites. In contrast, at higher pH levels, microbes tended to secrete more exopeptidases, increasing the formation of amino acids, dipeptides, etc.

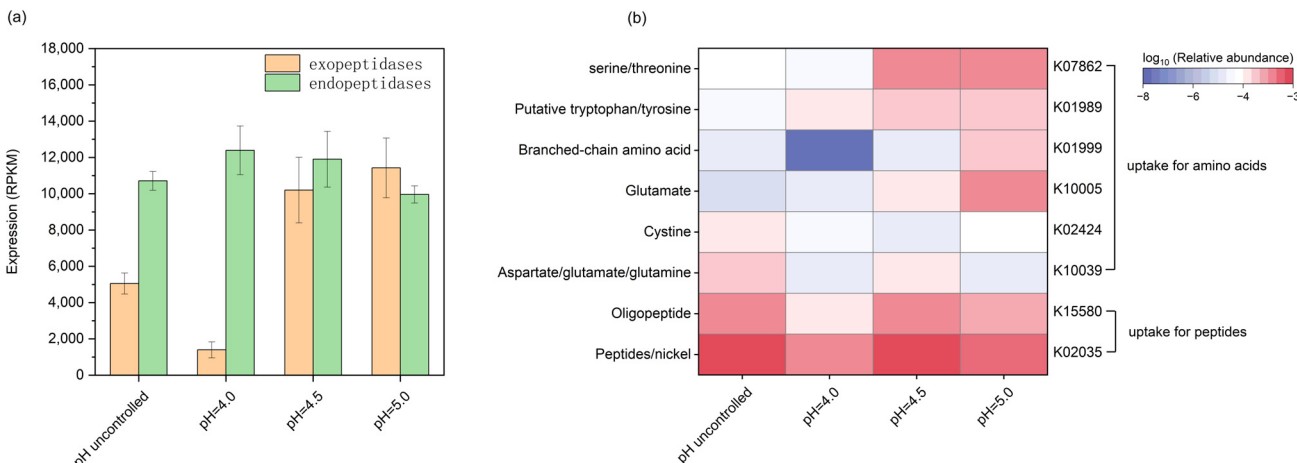

**Figure 4.** The transcriptional expression levels of peptidases and membrane transporters; (**a**) the relative expression of exopeptidases and endopeptidases; (**b**) the relative expression of some key orthologous genes associated with amino acids and peptides transportation systems.

The membrane transport is the second step in utilizing nitrogen-containing organic matter by microorganisms. Microorganisms can take up amino acids, oligopeptides, and polypeptides through two commonly used uptake systems: ATP-binding cassette (ABC) transporters and electrochemical potential-driven transporters [26]. Figure 4b displays the relative expression of some critical orthologous genes associated with these uptake systems, and the membrane transporters are here categorized into two groups based on the molecular size of the substrates. The relative expression of ABC transporters involved in shuttling peptides, specifically the oligopeptide transporter and peptides/nickel transporter, exhibited similar patterns. All the four reactors showed relatively high activity in the transport of peptides, and the activities showed no correlation with pH values ($p > 0.1$). The transporters for amino acids acted differently depending on the pH. Most work better at pH 4.5 and 5.0 than uncontrolled pH and pH 4.0. The gene that aids the ABC transporter in

bringing in glutamate (K10005) is much more active at higher pH levels (9.2 times at pH 4.5 and 931 times at pH 5.0 compared to pH 4.0). The result indicated that more glutamate was transported into the cell at high pH levels, likely enhancing amino acid metabolism because glutamate is an essential intermediary in aminotransferases [27].

After peptides and amino acids were taken, they were degraded further, and ammonium was released. Figure S2 illustrates the gene abundance related to specific metabolism based on the KEGG database and the expression levels of functional genes at the tertiary level involving amino acid metabolism. The functional genes associated with amino acid metabolism positively correlated with pH values (Spearman's correlation coefficient R2 = 0.863, $p < 0.001$) overall, while the correlation was different for specific ammonia acid. The metabolic pathways of amino acids are highly complex. Amino acids that are transported into cells undergo either catabolic or anabolic reactions, resulting in the release of ammonium or the synthesis of cell-building substances [29]. Amino acids can be degraded in pairs through the Stickland reaction or alone with the assistance of hydrogen-utilizing bacteria [31]. The Stickland reaction is the most common mode [32], in which a pair of amino acids act as an oxidative reactant and reductive reactant, respectively. The detailed metabolic pathways of amino acids were reconstructed with a focus on the pathways related to ammonium release (Figure 5). For simplification in the analysis, no new synthesis of amino acids was taken into account; instead, the amino acids were assumed to be broken down into intermediates within central metabolic pathways like glycolysis, the pentose phosphate pathway, and the TCA cycle [33], or they were converted into other intermediates and redirected towards other metabolic processes. Since nearly all the natural proteins are composed of L-amino acids [34], only L-amino acids were considered in the following discussion.

Glutamine, glutamate, asparagine, and aspartate can be directly converted into TCA cycle intermediates, such as 2-oxoglutarate and succinate. The pairing of glutamate and 2-oxoglutarate is central to transamination, serving as the first step in the metabolism of aspartate, alanine, and branched-chain amino acids [35]. 2-oxoglutarate acts as a universal acceptor of amino groups, transforming into glutamate. Notably, the expression of glutamate dehydrogenase (EC 1.4.1.4), catalyzing the oxidative deamination of glutamate with NADPH production [36], positively correlated with the fermentation pH values, implying an enhanced metabolic activity and greater ammonia release at high pH values. Aspartate transaminase (EC 2.6.1.1) also positively correlated with the pH values. The enzymes of the arginine deiminase (ADI) pathway, including arginine deiminase (EC 3.5.3.6), ornithine carbamoyl-transferase (EC 2.1.3.3), and carbamate kinase (EC 2.7.2.2), exhibited a negative correlation with pH values. Microbes utilize the ADI pathway at low pH values to protect against an acidic environment, as one mole of arginine can release two moles of ammonium via this pathway [37]. The metabolic pathways of alanine, cysteine, methionine, serine, glycine, and threonine are intertwined and interconnected with glycolysis and pyruvate metabolism. Ammonium was directly released during the breakdown of methionine, glycine, serine, and threonine. The relative expression levels of serine deaminase (EC 4.3.1.17) and threonine deaminase (EC 4.3.1.19) exhibited no significant difference across these reactors. However, the enzymes linked to methionine degradation, specifically methionine gamma-lyase (EC 4.4.1.11), and glycine degradation, namely aminomethyltransferase (EC 2.1.2.10), showed a strong positive correlation with pH values. The two enzymes also catalyze the removal of amino groups. Valine, leucine, and isoleucine, classified as branched-chain amino acids, undergo conversion into their respective α-ketoacids through the catalytic action of α-oxoglutarate-dependent aminotransferase. Eventually, they are transformed into isobutyrate, isovalerate, and 2-methylbutyrate, respectively [38]. At pH 4.5 and 5.0, α-oxoglutarate-dependent aminotransferase exhibited higher activity. This finding was consistent with the previous report on the emergence of branched-chain short-chain fatty acids in the fermentation with the pH increasing to 4.5 [39,40]. As for aromatic amino acids, phenylalanine could be converted to some intermediates and fur-

ther shunted to other metabolic pathways, while the metabolic pathways of tryptophan, tyrosine, and histidine are not complete (Figure S3).

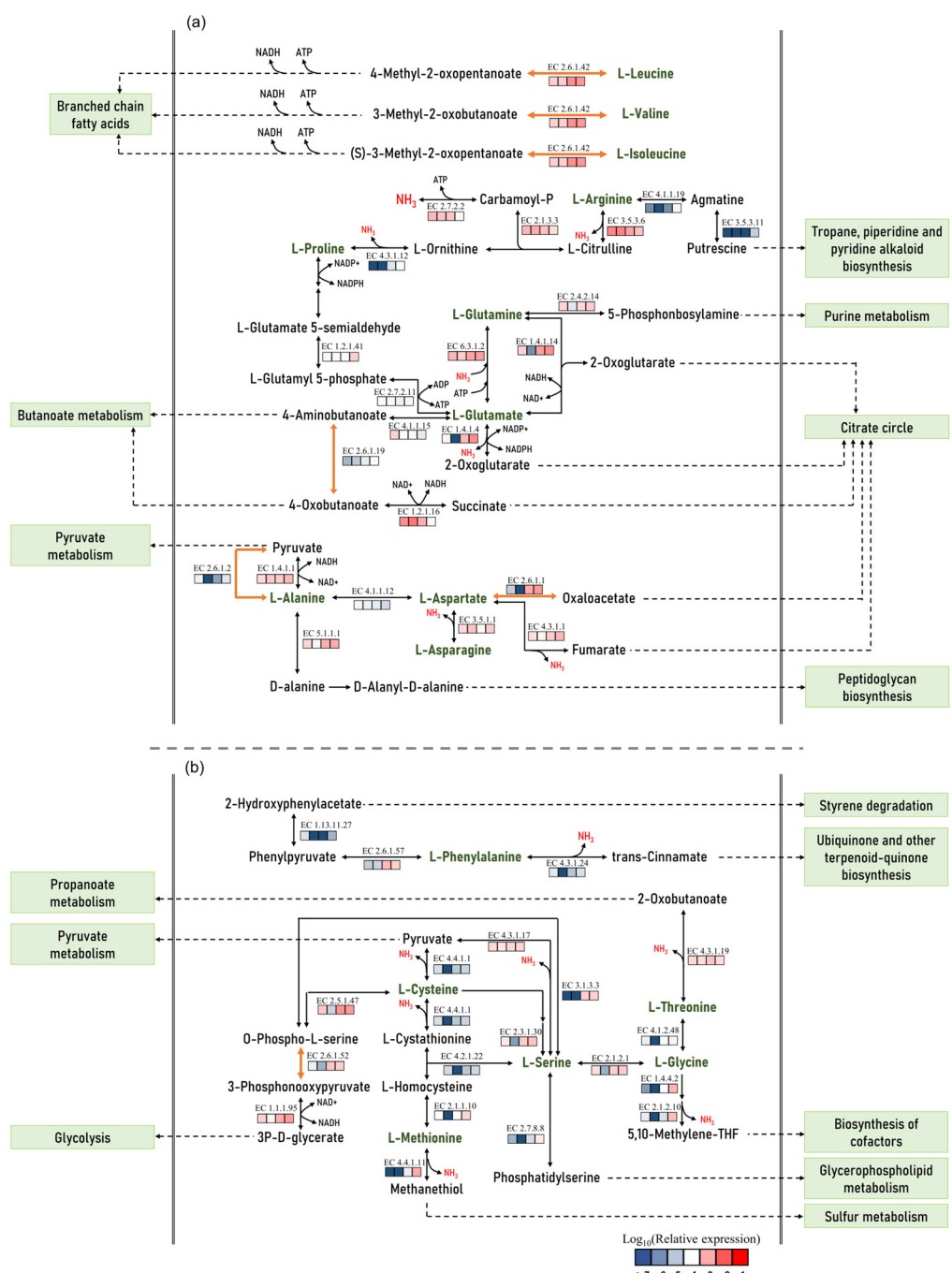

**Figure 5.** Metabolic pathways of amino acids along with their relative expression; (**a**) glutamate, glutamine, aspartate, asparagine, alanine, proline, arginine, leucine, valine, and isoleucine; (**b**) methionine, serine, cysteine, threonine, glycine, and phenylalanine; the orange arrows represent transamination reactions, with 2-oxoglutarate acting as the acceptor of the amino-group.

### 3.5. Key Genes and Their Carriers

At the community level, the activities of endopeptidases and the peptides uptake system were not correlated with the pH values. In contrast, the activity of exopeptidases and amino acid transportation significantly increased at high pH values. Arginine degradation via the ADI pathway was the primary source of ammonium release in the fermentation systems at pH 4.0–4.4. At pH 4.5 and 5.0, the community had much higher activity in

amino acid metabolism with ammonium release. Arginine deiminase (EC 3.5.3.6) and glutamate dehydrogenase (EC 1.4.1.4) were the critical enzymes for ammonia release at low pH and high pH, respectively. Sometimes, the taxa with the highest abundances may not necessarily be the primary contributors to community functions. To identify the most active microorganisms responsible for the expression of the two critical genes involved in ammonium release, mRNA sequences were mapped to the Metagenome-Assembled Genomes (MAGs) genomes. After undergoing quality control procedures, the metagenomic sequencing resulted in 231.4 Gb of paired-end data. Our metagenomic binning strategy successfully recovered 33 MAGs with high quality, indicating >70% completeness and <5% contamination. Figure 6 illustrates the relative contribution of MAGs to the expression of gene *arcA* (EC 3.5.3.6) and gene *gdhA* (EC 1.4.1.4).

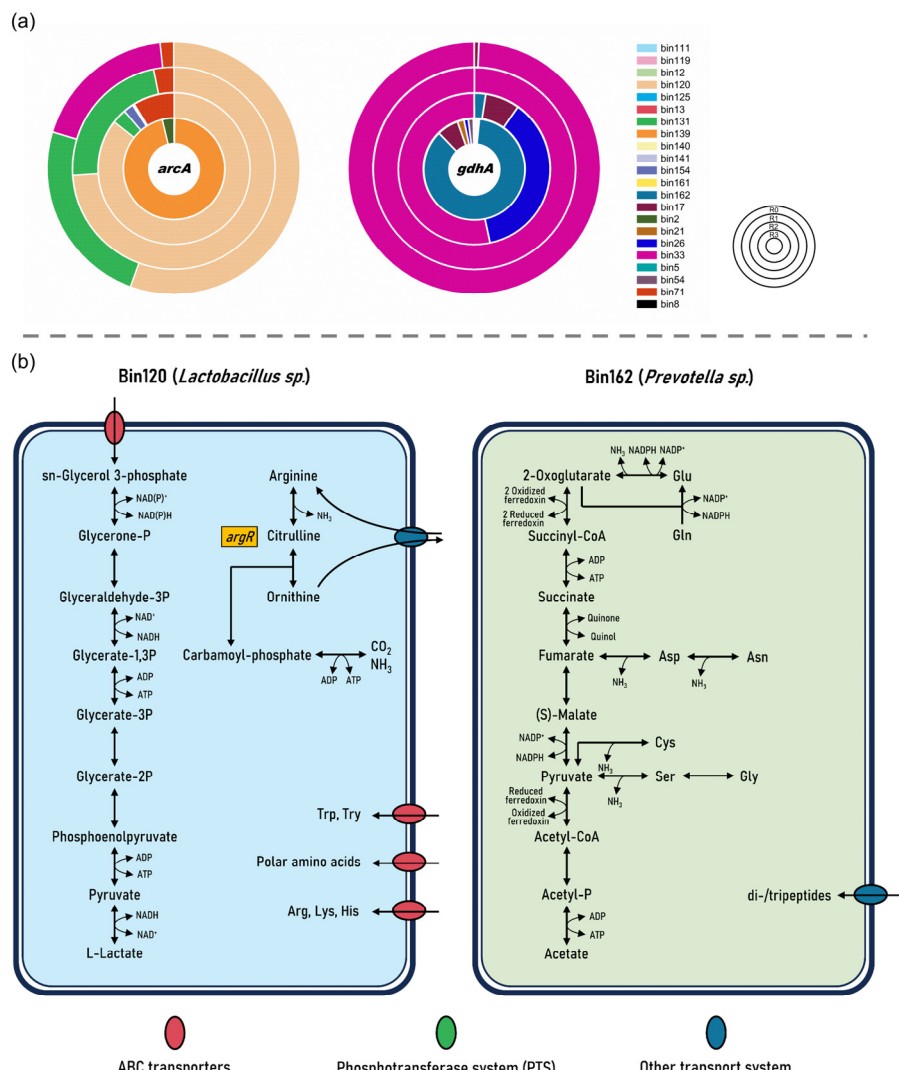

**Figure 6.** (**a**) The relative contribution of MAGs to transcripts of key genes related to ammonium release (gene *arcA* (arginine deiminase, EC 3.5.3.6); gene *gdhA* (glutamate dehydrogenase, EC 1.4.1.4); (**b**) pProposed metabolic pathways of Bin120 and Bin162.

Due to the similarity of microbial structures in R0, R1, and R2 (Figure 3), the relative contribution to gene *arcA* was similar in these reactors. Among these reactors, Bin120 (*Lactobacillus* sp.) exhibited the highest transcriptomic activity of gene *arcA*, contributing more than 50% of transcripts (55.6% in R0, 73.9% in R1, 85.6% in R2). Additionally, Bin131 (*Lactobacillus* sp.), Bin33 (*Coriobacteriaceae* sp.), and Bin71 (*Lactobacillus* sp.) also showed transcriptomic activity of *arcA*. As mentioned earlier, the gene *gdhA* expression level was positively correlated with the pH. In R3, most of the reads mapped to *gdhA* originated

from Bin162 (*Prevotella* sp.). At pH 4.0–4.5, Bin33 (*Coriobacteriaceae* sp.) played a more significant role in the transcript of gene *gdhA*, but its transcriptomic level was low. Bin120 (*Lactobacillus*) and Bin162 (*Prevotella*) were annotated, and 992 and 946 orthology genes were identified in Bin120 and Bin162, respectively. The metabolic pathway was analyzed using KEGG orthologous group ids (KO), as depicted in Figure 6.

Bin120 (completeness of 95.67%, contamination of 5.8%) belongs to the genus *Lactobacillus*. The metabolic reconstruction of Bin120 revealed incomplete glycolysis and pentose phosphate pathways because of the absence of key enzymes such as phosphofructokinase, aldolase, and transaldolase. Furthermore, the citrate cycle is also incomplete. As a result, Bin120 cannot utilize hexose for energy production like other LABs. Hence, no saccharide transporters are present within Bin120. However, Bin120 contains a complete gene set that encodes for an ATP-binding cassette (ABC) transporter of sn-glycerol 3-phosphate, an intermediate of lipid degradation [41]. This transporter allows the translocation of sn-glycerol 3-phosphate into the cell with ATP consumption. According to the KEGG annotation, once inside the cell, sn-glycerol 3-phosphate is converted to glycerone-3P by dehydrogenase and further catalyzed by isomerase to form glyceraldehyde-3P. Subsequently, glyceraldehyde-3P is directed into glycolysis pathways and converted to pyruvate, contributing to the complete pathway of the core module involving three-carbon compounds (glyceraldehyde-3P => pyruvate). Ultimately, pyruvate is converted to L-lactate, its final product, by L-lactate dehydrogenase. In the process of 1 mol sn-glycerol 3-phosphate transport and metabolism, 1 mol ATP and 1 mol NADPH are generated. Due to a lack of essential enzymes, Bin120 cannot synthesize other common acids and alcohols except lactic acid. Significantly, Bin120 houses the *arcABDC* gene cluster, which encodes for the ADI pathway and its regulator, ArgR. The ADI pathway comprises three enzymatic reactions facilitated by arginine deiminase (ArcA), ornithine carbamoyl-transferase (ArcB), and carbamate kinase (ArcC), facilitating the conversion of arginine into ornithine. The end product, ornithine, and extracellular arginine are exchanged via the putative arginine-ornithine antiporter (ArgD). The whole ADI system is activated by ArgR [42]. In this process, the conversion of 1 mol arginine can release 2 mol $NH_3$, which raises the internal pH and enhances the cell's tolerance to an acidic environment [37]. Additionally, this process also results in the generation of energy (1 mol ATP). Furthermore, Bin120 houses gene sets encoding other amino acid transporters, such as the tryptophan/tyrosine transporter (K01989, K05832, K05832), the polar amino acid transporter (K02028, K02029, K02030), and the arginine/lysine/histidine transporter (K23059, K17077, K23060), all of which belong to the ABC transporters. Bin120 employs the transported amino acids for protein synthesis. Despite the relatively low expression levels of Bin120 (proportions of metatranscriptomic reads mapped to Bin120 from the total reads were only $2.32 \pm 0.83‰$ and $2.22 \pm 0.02‰$ in R0 and R1, respectively), this organism plays a pivotal role in ammonium release during the fermentation at low pH. The analysis hinted that lipids in the substrate might stimulate the proliferation of certain *Lactobacillus* species, which in turn could boost ammonium release. Hence, the removal or recovery of lipids prior to FW fermentation may be an effective strategy to mitigate ammonium release.

Bin162, characterized by a completeness of 81.35% and a contamination of 2.31%, is attributed to the genus *Prevotella* and had a relative expression of $3.4 \pm 1.0\%$ in R3. An analysis of Bin162 genome revealed that the glycolysis pathway, the pentose phosphate pathway, and the citrate cycle were incomplete. Moreover, no complete gene clusters of ABC transporters and phosphotransferases were found. Interestingly, the genome does contain a di-/tripeptides transporter (K03305), a proton-dependent oligopeptide transporter (POT) that relies on the electric potential difference across the cell membrane to uptake di- and tripeptides [43]. The metabolic pathways for energy and material were reconstructed using annotated genes. Bin162 uptakes oligopeptides via POT and converts them into intermediates of the citrate cycle and glycolysis, which are subsequently converted to acetate. Specifically, glutamate and glutamine are converted to 2-oxoglutarate, while asparagine and aspartate are transformed into fumarate. These compounds get involved in

the citrate cycle and glycolysis, generating pyruvate. Besides, glycine, serine, and cysteine can be converted to pyruvate by a series of enzymes, including transaminase, ammonia-lyase, etc. Eventually, pyruvate undergoes a three-step process to convert into acetate, which is released into the extracellular space. In this process, Bin162 obtains the required energy and substances. Due to the lack of key enzymes, other organic acids and alcohols could not be synthesized. Oligopeptides are probably the primary source of energy and substance for Bin162. Consequently, this microorganism revealed high activity of amino acid metabolism. During amino acid degradation or conversion, amino groups of some amino acids (such as aspartate, phenylalanine, etc.) were transformed to the transfer station glutamate. This result explains the high expression level of gene *gdhA*, which encodes the enzyme responsible for removing amino from glutamate.

## 4. Conclusions

FW fermentation exhibited different types at pH 4.0–5.0. The pH increase did not enhance the hydrolysis of solid proteins but promoted the conversion from soluble nitrogen-containing substances to ammonium. At pH 4.0–4.5, the activity of amino acid metabolism was relatively low, with most of the ammonium resulting from the ADI pathway. A species of *Lactobacillus* (Bin120), which utilizes sn-glycerol 3-phosphate to form lactic acid, appeared to play a crucial role in ammonium release. Conversely, at pH 5.0, various protein-to-ammonium conversion pathways were enhanced, and a species of *Prevotella* (Bin62) was found to be key in ammonia release.

**Supplementary Materials:** The following supporting information can be downloaded at: https://www.mdpi.com/article/10.3390/fermentation10030129/s1, Metagenomic analyses, Metatranscriptomic analyses; Figure S1: Quantities of bacteria in reactors using qPCR; Figure S2: Relative gene expression of representative KEGG functional categories; Figure S3: Tryptophan metabolism pathway (ko00380) (a), tyrosine metabolism pathway (ko00350) (b), and histidine metabolism pathway (ko00340) (c) in FW fermentation reactors. The detected and expressed enzymes are colored pink.

**Author Contributions:** Conceptualization, H.L.; methodology, C.Z. and L.Y.; validation, C.Z.; formal analysis, C.Z. and H.L.; investigation, C.Z. and L.Y.; resources, Z.D.; data curation, C.Z.; writing—original draft, C.Z.; writing—review & editing, C.Z., L.Y. and H.L.; visualization, C.Z.; supervision, H.L. and Z.D.; project administration, H.L and Z.D. All authors have read and agreed to the published version of the manuscript.

**Funding:** This research was funded by the Shenzhen Science and Technology Innovation Commission (KCXFZ20211020163556020).

**Institutional Review Board Statement:** Not applicable.

**Informed Consent Statement:** Not applicable.

**Data Availability Statement:** Dataset available on request from the authors.

**Conflicts of Interest:** The authors declare no conflict of interest.

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
