# Peer review of "Nitrogen Metabolism during Anaerobic Fermentation of Actual Food Waste under Different pH Conditions"

_fermentation, doi:10.3390/fermentation10030129_

Round 1

Reviewer 1 Report

Comments and Suggestions for Authors

Find the specific comment for the authors for more details.

 1.     How many replicates were used in each treatment?

2.     The genus and species names of bacteria throughout the manuscript should be consistently in italics.

3.     Line 74-78, what is the novelty of this study?

4.     The abstract should be revised to add some specific quantitative data and highlight results more interestingly providing major key findings.

5.     The introduction didn’t establish the background of the problem studied. It should be rewritten in a precise manner.

6.     In Table 1 Total Kjeldahl nitrogen (TKN) and Soluble Kjeldahl nitrogen (SKN) value need to provide the units.

7.     Page 2, Line 80-81, The substrate and the inoculum used in the fermentation experiments were collected from a treatment plant in Shenzhen, China. Authors need to provide the food waste composition.

8.     Page 4, Line 150-151- “The inoculum, the substrate, and the fermentation broth samples were first centrifuged”. Why do you want to centrifuge the inoculum and substrate separated? How was it done? Explain clearly.

9.     In Lines 139-140, “Quantitative PCR (qPCR) analysis was performed using an ABI7300 instrument (Ap-139 plied Biosystems, USA). ChamQ SYBR Color qPCR Master Mix (2X) (Vazyme, China) was 140 used with the 341F and 806R mix primer sets”. In addition, the used primer was not suitable for archaea (methanogens) analysis and methanogenic genes prediction.

10.  In Figure 1a, the right site y-axis pH value should be included.

11.  In terms of N transformation, authors analyzed functional genes which was really good, but there is no physiochemical data regarding NH4+-N and, TOC, H2, CH4, and CO2 concentrations during the anaerobic fermentation. Please provide this data, otherwise, the relatives of functional genes are meaningless. Microbial data can't be used to explain/match the physical phenomena.

12.  Please provide the error bar in Figures 1a,b, and d.

13.  Line 321-322- Please correct the form ex-opeptidases.

14.  What was the substrate and inoculum loading rate (VS basis)?

Author Response

Response to Comments

We sincerely thank the reviewers for the time spent evaluating our manuscript and for their helpful comments. We addressed all comments when revising the manuscript, as described below. We also highlighted the changes in yellow in the revised version.

Comment 1

How many replicates were used in each treatment?

Reply:

In this study, the fermentation experiments were conducted in four continuous reactors (pH uncontrolled, pH 4.0, pH 4.5, and pH 5.0), rather than batch reactors. Every day, fermentation broth was discharged and then new FW was fed once. Hence, the same operation condition was repeated many times in a long period. Such a way is usually applied in continuous biochemical experiments (Gu et al., 2018; Han et al., 2018; Kim et al., 2016; Li et al., 2021; Wang et al., 2013; Yang et al., 2023; Zhang et al., 2017; Zhang et al., 2020a; Zhang et al., 2020b). For any sample collected from the reactors during the steady state, the measurement (for example, ammonia nitrogen, pH, etc.) was repeated three times, and the average was used for analyses.

Comment 2

The genus and species names of bacteria throughout the manuscript should be consistently in italics.

Reply:

We acknowledge our oversight during the submission process. We have now rectified this issue by ensuring that all the genus and species names of bacteria in the manuscript are consistently presented in italics.

Comment 3

Line 74-78, what is the novelty of this study?

Reply:

We clarify the contribution of this work in the fourth paragraph in the introduction section. This is the first comprehensive investigation of nitrogen conversion and distribution during food waste fermentation under different pH conditions. Specifically, our study focuses on elucidating the nitrogen distribution between the solid phase and liquid phase. We have incorporated this information into the introduction section of the article to better highlight the unique contribution of our study.

Comment 4

The abstract should be revised to add some specific quantitative data and highlight results more interestingly providing major key findings.

Reply:

We appreciate the valuable feedback provided by the reviewer. We have revised the abstract as per your suggestions.

Acidogenic fermentation can convert food waste (FW) into small molecules of acids and alcohols, and the broth can be used as the carbon source of denitrification in wastewater treatment plants. However, the soluble nitrogen-containing substances generated in fermentation influence the quality of the carbon source, and microbial nitrogen transformation under different pH conditions has rarely been reported. In this study, four FW fermentation systems were operated continuously with a focus on nitrogen transformation, and metagenomic and metatranscriptomic analyses were used to reveal the metabolic pathways. The results showed that approximately 70% of nitrogen existed in solid organic matter, and the dissolution of solid proteins was limited at pH 4.0–5.0. The concentration of soluble nitrogen, encompassing both soluble organic nitrogen and ammonium, remained relatively stable across various pH conditions. However, high pH values promoted the conversion of soluble nitrogen-containing substances to ammonium, and its concentration in-creased by 122%, 180%, 202%, and 267% at pH 4.00, pH 4.27, pH 4.50, pH 5.00. Lactobacillus played a crucial role in ammonium production via the arginine deiminase pathway at pH 4.0–4.5, and Prevotella was the key contributor with the assistance of glutamate dehydrogenase at pH 5.0. The findings provide insights into organic nitrogen transformation in acidogenic fermentation for optimizing FW treatment processes.

Comment 5

The introduction didn’t establish the background of the problem studied. It should be rewritten in a precise manner.

Reply:

We rewrote the paragraphs in the introduction section to clarify the background. We explained the reason of controlling nitrogen content in FW fermentation broth before it is taken as a carbon source.

…The fermentation liquid containing acetic acid, propionic acid, ethanol, etc. can be used as the carbon source for biological denitrification in wastewater treatment plants (WWTPs) [8, 9]. Methanol, ethanol, and glucose are the typical commercial carbon sources [8]. The combination of FW fermentation and wastewater treatment provides a new pathway for FW utilization with high efficiency and low cost. Also, it reduces the economic and environmental burden of WWTPs by avoiding the usage of fossil carbon sources [1]. However, the presence of soluble nitrogen-containing compounds influences the quality of fermentation liquid as a carbon source [10-12] and consequently challenges the standard discharge of WWTPs. Therefore, this new pathway of FW utilization must disclose the nitrogen transformation first and finally reduce the dissolution of solid nitrogen by controlling FW fermentation conditions.…

Comment 6

In Table 1 Total Kjeldahl nitrogen (TKN) and Soluble Kjeldahl nitrogen (SKN) value need to provide the units.

Reply:

Thanks for your reminder. We have now included the units for Total Kjeldahl nitrogen (TKN) and Soluble Kjeldahl nitrogen (SKN) in Table 1.

Comment 7

Page 2, Line 80-81, The substrate and the inoculum used in the fermentation experiments were collected from a treatment plant in Shenzhen, China. Authors need to provide the food waste composition.

Reply:

The food waste treated by the plant is sourced from restaurants and households. It comprises waste staples, vegetables, meat, and other constituents. Detailed physiochemical characteristics of the food waste materials employed in our experiments can be found in Table 1.

Comment 8

Page 4, Line 150-151- “The inoculum, the substrate, and the fermentation broth samples were first centrifuged”. Why do you want to centrifuge the inoculum and substrate separated? How was it done? Explain clearly.

Reply:

There may have been some confusion due to our unclear expression in this section. In this section, we introduced the method of sample analysis. We separately analyzed the substrate (before adding it to the reactors), inoculum (before adding it to the reactors), and the fermentation broth (taken from the effluent of the reactors). Each sample was individually centrifuged and filtered to measure the soluble parameters. To avoid any further misunderstanding, we have revised the sentence to read as follows: “In order to analyze the soluble parameters, the samples were first centrifuged at 5800 g for 5 minutes, followed by filtration of the resulting supernatant through a 0.45 μm filtration membrane.”

Comment 9

In Lines 139-140, “Quantitative PCR (qPCR) analysis was performed using an ABI7300 instrument (Ap-139 plied Biosystems, USA). ChamQ SYBR Color qPCR Master Mix (2X) (Vazyme, China) was 140 used with the 341F and 806R mix primer sets”. In addition, the used primer was not suitable for archaea (methanogens) analysis and methanogenic genes prediction.

Reply:

Thank you for your professional feedback. We appreciate your concerns regarding the primer pair 341F and 806R, which is commonly used for bacterial analysis. In acidogenic fermentation systems, the predominant microorganisms are indeed bacteria, and the abundance of archaea is extremely low. In our study, the number of reads mapped to the Archaea domain was only 299,606, whereas the number of reads mapped to the Bacteria domain was a substantial 332,707,842. The archaeal reads account for less than 1‰ of the bacterial reads. Several factors contribute to this low abundance of archaea (specifically methanogens):

  1. The pH levels in our acidogenic fermentation system (ranging from 4.0 to 5.0) are not conducive to the survival of methanogens, as these microorganisms thrive under more neutral or alkaline conditions (Lee et al., 2010).
  2. The relatively short hydraulic retention time (HRT) of 4 days in our study does not allow for the accumulation of methanogens. The rapid turnover of the system is likely to wash out archaea (Kim et al., 2006).

Given the predominance of bacteria in our reactors, we utilized qPCR to quantify the microbial populations, with a focus on bacteria. Consequently, the primer pair 341F and 806R was chosen to target our intended microbial population effectively. To address the confusion and provide further clarification, we have incorporated additional relevant information into the manuscript.

Comment 10

In Figure 1a, the right site y-axis pH value should be included.

Reply:

Fig. 1a has the right-site y-axis pH value. However, we did not include it in Figs. 1b-d because the pH values of R1, R2, and R3 were maintained at 4.00±0.04, 4.50±0.04, and 5.00±0.04, respectively, by an automatic pH controller, as described in Lines 104-107 (the manuscript for the first review). This information has been clearly described in Lines 106-107.

Comment 11

In terms of N transformation, authors analyzed functional genes which was really good, but there is no physiochemical data regarding NH4+-N and, TOC, H2, CH4, and CO2 concentrations during the anaerobic fermentation. Please provide this data, otherwise, the relatives of functional genes are meaningless. Microbial data can't be used to explain/match the physical phenomena.

Reply:

In our paper, we initially conducted an extensive analysis of nitrogen distribution and conversion during fermentation under various pH conditions. In Section 3.2, we provided detailed information on nitrogen concentrations in different forms, including ammonium (NH4+-N), soluble organic nitrogen, and insoluble organic nitrogen, etc. Our findings revealed that an increase in pH did not significantly enhance the hydrolysis of solid proteins but did promote the conversion from soluble nitrogen-containing compounds to ammonium. The conversion of organic carbon can be found in Fig. 1, which describes the concentrations of carbon-containing substances. The gas emissions were neglectable at low pH, and the results can be found in our previous study(Yang et al., 2022; Yang et al., 2023).

Comment 12

Please provide the error bar in Figures 1a,b, and d.

Reply:

In this study, the fermentation experiments were conducted in four continuous reactors (pH uncontrolled, pH 4.0, pH 4.5, and pH 5.0), rather than batch reactors. Every day, fermentation broth was discharged and then new FW was fed once. Hence, the same operation condition was repeated many times in a long period. Such a way is usually applied in continuous biochemical experiments. When evaluating the performance, we utilized the average of the data in a stable period more than 8 days (equivalent to two times the HRT). Similar methods have also been employed in the literature.

Comment 13

Line 321-322- Please correct the form ex-opeptidases.

Reply:

Thanks for bringing this to our attention. We have made the necessary correction

Comment 14

What was the substrate and inoculum loading rate (VS basis)?

Reply:

The inoculum (10%, v/v) was added only once at the beginning of the experiment, as described in Lines 110-111 (the manuscript for the first-time review). During the over 80-day semi-continuous reaction, the substrate was added daily after discharging a certain quantity of fermented broth. The organic loading rate (OLR) ranged from 23 to 36 g VS/(L·d), as described in Line 115 (the manuscript for the first review).

References

Gu, X.Y., Liu, J.Z., Wong, J.W.C., 2018. Control of lactic acid production during hydrolysis and acidogenesis of food waste. Bioresour. Technol. 247, 711–715.

Han, W., He, P., Shao, L., Lü, F., 2018. Metabolic Interactions of a Chain Elongation Microbiome. Applied and environmental microbiology 84 (22).

Kim, J.K., Oh, B.R., Chun, Y.N., Kim, S.W., 2006. Effects of Temperature and Hydraulic Retention Time on Anaerobic Digestion of Food Waste. Journal of bioscience and bioengineering 102 (4), 328–332.

Kim, M.-S., Na, J.-G., Lee, M.-K., Ryu, H., Chang, Y.-K., Triolo, J.M., Yun, Y.-M., Kim, D.-H., 2016. More value from food waste: Lactic acid and biogas recovery. Water Res. 96, 208–216.

Lee, D., Ebie, Y., Xu, K., Li, Y.-Y., Inamori, Y., 2010. Continuous H2 and CH4 production from high-solid food waste in the two-stage thermophilic fermentation process with the recirculation of digester sludge. Bioresour. Technol. 101 Suppl 1, S42-7.

Li, W., Gao, J., Zhuang, J.-L., Yao, G.-J., Zhang, X., Liu, Y., Liu, Q.-K., Shapleigh, J.P., Ma, L., 2021. Metagenomics and metatranscriptomics uncover the microbial community associated with high S0 production in a denitrifying desulfurization granular sludge reactor. Water research 203, 117505.

Wang, B., Li, Y., Ren, N., 2013. Biohydrogen from molasses with ethanol-type fermentation: Effect of hydraulic retention time. International Journal of Hydrogen Energy 38 (11), 4361–4367.

Yang, L., Chen, L., Zhao, C., Li, H., Cai, J., Deng, Z., Liu, M., 2023. Biogas slurry recirculation regulates food waste fermentation: Effects and mechanisms. Journal of Environmental Management 347, 119101.

Yang, L., Chen, L., Li, H., Deng, Z., Liu, J. 2022. Lactic acid production from mesophilic and thermophilic fermentation of food waste at different pH. J Environ Manage, 304, 114312.

Zhang, S., Liu, M., Chen, Y., Pan, Y.-T., 2017. Achieving ethanol-type fermentation for hydrogen production in a granular sludge system by aeration. Bioresour. Technol. 224, 349–357.

Zhang, W., Xu, X., Yu, P., Zuo, P., He, Y., Chen, H., Liu, Y., Xue, G., Li, X., Alvarez, P.J.J., 2020a. Ammonium Enhances Food Waste Fermentation to High-Value Optically Active l -Lactic acid. ACS Sustainable Chem. Eng. 8 (1), 669–677.

Zhang, Z., Zhang, Y., Chen, Y., 2020b. Comparative Metagenomic and Metatranscriptomic Analyses Reveal the Functional Species and Metabolic Characteristics of an Enriched Denitratation Community. Environmental science & technology 54 (22), 14312–14321.

Reviewer 2 Report

Comments and Suggestions for Authors

The work is based on the study of the behavior of microorganisms against different conditions in anaerobic digestion focused on a process of organic acid formation by controlling the pH of each reactor. The methods used are correct and could be used to reach the conclusions if the tests had been well designed, but this does not seem to have been the case.

Line 18 in the abstract "twofold" is separate.

Line 32 lacks one of the most common products obtained from FW, such as energy by combustion or even bioethanol through aerobic digestion.

Line 34. Anerobic digestion is not the most used system, it may be in some countries, but not worldwide, so the authors should specify it better.

Because the substrate has been taken every so often from the treatment plant instead of taking a larger amount at the beginning in order to develop the whole trial with the same substrate, avoiding variability of composition, which can lead to variations in metabolic pathways, and in the microbiotic profile.

Line 165, write the formula KH2PO4 correctly.

Line 171, specify the time in days as days and not as d.

In Figure 1, the pH value should be indicated in the reactors where the pH has been set in order to facilitate the understanding.

The design of the tests as well as the analytical methods used have been correct, except for the number of reactors used, no replicates have been used, which does not allow generating reliable results from tests with microorganisms. The normal practice in this type of assay is to use between three or even four reactors. Not having used replicates is a very negative aspect that does not allow establishing statistical differences between the results. For example, the great difference found in acid formation between reactors should be due external issues that could be detected using replicates.

On the other hand, although the methods for the determination of the microbial population are correct, when the feed varies with time, it is not possible to observe which microorganisms come from the feed, how they vary with the change of feed, nor those that have been promoted by the anaerobic digestion. This is why the results are also not as reliable because it is not possible to distinguish between changes due to feeding and those due to fermentation.

The same occurs with the amino acid content, since the input and output compositions are not compared, it is not possible to be sure which amino acids come from the raw material and which come from the anaerobic digestion.

Author Response

Response to Comments

We sincerely thank the reviewers for the time spent evaluating our manuscript and for their helpful comments. We addressed all comments when revising the manuscript, as described below. We also highlighted the changes in yellow in the revised version.

Comment 1

Line 18 in the abstract "twofold" is separate.

Reply:

Thank you for bringing this to our attention. We have made the necessary revision in the abstract.

Comment 2

Line 32 lacks one of the most common products obtained from FW, such as energy by combustion or even bioethanol through aerobic digestion.

Reply:

We have revised the original sentence and included the information.

Comment 3

Line 34. Anerobic digestion is not the most used system, it may be in some countries, but not worldwide, so the authors should specify it better.

Reply:

We revised the sentence to “Anaerobic digestion is a commonly-used technology”.

Comment 4

Because the substrate has been taken every so often from the treatment plant instead of taking a larger amount at the beginning in order to develop the whole trial with the same substrate, avoiding variability of composition, which can lead to variations in metabolic pathways, and in the microbiotic profile.

Reply:

In this plant, a large buffer storage tank can ensure the stabilization of the feedstock to fermentation tanks. Certainly, this issue is also critical for the plant to operate smoothly. In fact, the substrate was relatively stable, and the characteristics were recorded in Table 1. In addition, for a long period (more than 80 days), you can not ensure the samples keep unchanged even in a refrigerator, and cryopreservation can also damage the structure of the FW samples. Therefore, we decided to collect substrate every two weeks, and measured its characteristics before usage. Thus, we can ensure the consistency of substrate. A slight fluctuation can not influence the performance of such continuous reactors, as illustrated in Figure 1.

Comment 5

Line 165, write the formula KH2PO4 correctly.

Reply:

Thank you for pointing that out. We have made the necessary correction.

Comment 6

Line 171, specify the time in days as days and not as d.

Reply:

Thanks for your feedback. We have corrected it.

Comment 7

The design of the tests as well as the analytical methods used have been correct, except for the number of reactors used, no replicates have been used, which does not allow generating reliable results from tests with microorganisms. The normal practice in this type of assay is to use between three or even four reactors. Not having used replicates is a very negative aspect that does not allow establishing statistical differences between the results. For example, the great difference found in acid formation between reactors should be due external issues that could be detected using replicates

Reply:

We completely understand the importance of replicates in batch assays. However, in this study we operated continuous reactors, implying that all the experiments were repeated every day in a long period. So, the evaluation was conducted based on many replicated results. In fact, single continuous biochemical reactor is commonly used in organic waste treatment studies, for example,

  1. In the study by Tian et al. (2023), food waste fermentation liquid was used as a supplementary carbon source for a bench-scale step-feed three-stage anoxic/aerobic system (SFTS-A/O). The SFTS-A/O was conducted continuously with one reactor to investigate nutrient removal and microbial community responses.
  2. Le et al. (2022) tested a two-stage anaerobic membrane bioreactor for co-treatment of food waste and kitchen wastewater for biogas production and nutrient recovery. The experiment was conducted continuously for a total of 160 days, with each condition lasting 50-60 days. The study used one replication.
  3. Crognale et al. (2021) employed metagenomics to investigate single-stage fermentation processes for the production of medium-chain fatty acids in a semi-continuous reactor treating the extract of real food waste. Two sequential acidogenic fermentation tests were carried out in continuous mode, with one replication for each condition.
  4. Li et al. (2021) employed meta-omics to describes the microbial mechanism in an efficient denitrification S0 production bioreactor based on inoculation with anaerobic granular sludge. Their reactor was conducted continuously for about 170 days, and only one reactor was used.
  5. Zhao et al. (2021), operated three independent membrane biofilm reactors continuously for over 600 days, with one replication for each reactor, to study the impact of feeding rate of electron acceptors on methane bioconversion to short-chain fatty acids.
  6. Ma et al. (2020) set up four groups for semi-continuous 150-days experiment to explore the effect of liquid digestate recirculation on the food waste ethanol-type anaerobic digestion system. For each experimental group, only one replication was used.
  7. Cheng et al. (2020) investigated the anaerobic treatment of food waste at different high solid concentrations. They operated a single reactor for 180 days with one replication.
  8. Zhang et al. (2020) conducted a 270-day study using a lab-scale sequencing batch reactor with one replication to reveal the taxonomic composition and gene expression patterns of the denitrification community enriched from activated sludge.
  9. Roghair et al. (2018) investigated the quantitative effect of CO2 loading rate on ethanol usages in a chain elongation process. In their study, a continuous reactor was operated with one replication.
  10. The study by Han et al. (2018) provides insights into the microbial diversity and predictive microbial metabolic pathways of a mixed-culture carbon chain elongation microbiome on the basis of a comparative analysis of the metagenome and metatranscriptome. In their study, a 21L upflow blanket filter reactor was operated for over 70 days with only one replication.

Comment 8

On the other hand, although the methods for the determination of the microbial population are correct, when the feed varies with time, it is not possible to observe which microorganisms come from the feed, how they vary with the change of feed, nor those that have been promoted by the anaerobic digestion. This is why the results are also not as reliable because it is not possible to distinguish between changes due to feeding and those due to fermentation.

Reply:

In a stable continuous fermentation reactor, the microorganisms in feed (FW) did not damage the indigenous microbial structure, while the performance was indeed influenced significantly by pH (environmental condition). In our previous study, we had explained the shock effects of feed on the native microbes(Yang et al., 2023). The short-term effects were quickly smoothed out. This is also identical with the real situation in a treatment plant. In this study, we focused on the N metabolism along with the evolution of microbes under different environmental conditions, and the feed itself did not interrupt our investigation on this issue.

Comment 9

The same occurs with the amino acid content, since the input and output compositions are not compared, it is not possible to be sure which amino acids come from the raw material and which come from the anaerobic digestion.

Reply:

In anaerobic metabolism, the de novo synthesis of amino acids was relatively limited, and accordingly, the amino acids embedded in the proteins of FW (>10000 mg/L) were evaluated, as described in Line 356-359 (the manuscript for the first-round review).

References

Cheng, H., Li, Y., Kato, H., Li, Y.-Y., 2020. Enhancement of sustainable flux by optimizing filtration mode of a high-solid anaerobic membrane bioreactor during long-term continuous treatment of food waste. Water research 168, 115195.

Crognale, S., Braguglia, C.M., Gallipoli, A., Gianico, A., Rossetti, S., Montecchio, D., 2021. Direct Conversion of Food Waste Extract into Caproate: Metagenomics Assessment of Chain Elongation Process. Microorganisms 9 (2).

Le, T.-S., Nguyen, P.-D., Ngo, H.H., Bui, X.-T., Dang, B.-T., Diels, L., Bui, H.-H., Nguyen, M.-T., Le Quang, D.-T., 2022. Two-stage anaerobic membrane bioreactor for co-treatment of food waste and kitchen wastewater for biogas production and nutrients recovery. Chemosphere 309 (Pt 1), 136537.

Li, W., Gao, J., Zhuang, J.-L., Yao, G.-J., Zhang, X., Liu, Y., Liu, Q.-K., Shapleigh, J.P., Ma, L., 2021. Metagenomics and metatranscriptomics uncover the microbial community associated with high S0 production in a denitrifying desulfurization granular sludge reactor. Water research 203, 117505.

Ma, X., Yu, M., Yang, M., Zhang, S., Gao, M., Wu, C., Wang, Q., 2020. Effect of liquid digestate recirculation on the ethanol-type two-phase semi-continuous anaerobic digestion system of food waste. Bioresour. Technol. 313, 123534.

Roghair, M., Hoogstad, T., Strik, D.P.B.T.B., Plugge, C.M., Timmers, P.H.A., Weusthuis, R.A., Bruins, M.E., Buisman, C.J.N., 2018. Controlling Ethanol Use in Chain Elongation by CO2 Loading Rate. Environmental science & technology 52 (3), 1496–1505.

Tian, Z., Li, G., Xiong, Y., Cao, X., Pang, H., Tang, W., Liu, Y., Bai, M., Zhu, Q., Du, C., Li, M., Zhang, L., 2023. Step-feeding food waste fermentation liquid as supplementary carbon source for low C/N municipal wastewater treatment: Bench scale performance and response of microbial community. Journal of Environmental Management 345, 118434.

Yang, L., Chen, L., Zhao, C., Li, H., Cai, J., Deng, Z., Liu, M., 2023. Biogas slurry recirculation regulates food waste fermentation: Effects and mechanisms. Journal of Environmental Management 347, 119101.

Zhang, Z., Zhang, Y., Chen, Y., 2020. Comparative Metagenomic and Metatranscriptomic Analyses Reveal the Functional Species and Metabolic Characteristics of an Enriched Denitratation Community. Environmental science & technology 54 (22), 14312–14321.

Zhao, L., Chen, H., Yuan, Z., Guo, J., 2021. Interactions of functional microorganisms and their contributions to methane bioconversion to short-chain fatty acids. Water research 199, 117184.

Round 2

Reviewer 1 Report

Comments and Suggestions for Authors

The Authors addressing most of the reviewer comments

Reviewer 2 Report

Comments and Suggestions for Authors

All issues raised in the first review have been well answered by the authors and the changes have been properly implemented.